# Diagnostic Value of EUS-Guided Fine-Needle Aspiration Biopsy for Gastric Linitis Plastica with Negative Endoscopic Biopsy

**DOI:** 10.3390/jcm10163716

**Published:** 2021-08-20

**Authors:** Ryutaro Takada, Kosuke Minaga, Akane Hara, Yasuo Otsuka, Shunsuke Omoto, Ken Kamata, Kentaro Yamao, Mamoru Takenaka, Satoru Hagiwara, Hajime Honjo, Shigenaga Matsui, Takaaki Chikugo, Tomohiro Watanabe, Masatoshi Kudo

**Affiliations:** 1Department of Gastroenterology and Hepatology, Kindai University Faculty of Medicine, Osaka-Sayama 589-8511, Japan; ryutaro1101@med.kindai.ac.jp (R.T.); akanehara@med.kindai.ac.jp (A.H.); yotsuka624@gmail.com (Y.O.); shunsuke.oomoto@gmail.com (S.O.); ky11@leto.eonet.ne.jp (K.K.); k-yamao@med.kindai.ac.jp (K.Y.); matakenaka@med.kindai.ac.jp (M.T.); hagi-318@hotmail.co.jp (S.H.); hnjgel@gmail.com (H.H.); ma2i@med.kindai.ac.jp (S.M.); tomohiro@med.kindai.ac.jp (T.W.); m-kudo@med.kindai.ac.jp (M.K.); 2Department of Diagnostic Pathology, Kindai University Faculty of Medicine, Osaka-Sayama 589-8511, Japan; tchikugo@mac.com

**Keywords:** biopsy, endoscopic ultrasound, fine-needle aspiration, gastric linitis plastica, gastric cancer, linitis plastica

## Abstract

Due to the tendency of gastric linitis plastica (GLP) to cause extensive submucosal infiltration, a superficial endoscopic biopsy sometimes yields no evidence of malignancy, hindering definite diagnosis. The present study was a single-center retrospective analysis of 54 consecutive patients diagnosed with GLP between 2016 and 2020 to evaluate EUS-guided fine-needle aspiration (EUS-FNA) biopsy outcomes in patients with negative endoscopic biopsy findings. A pathological GLP diagnosis was achieved by endoscopic biopsy in 40 patients (74.1%). EUS-FNA biopsy with a 22-gauge needle was performed in 13 of the remaining 14 patients, and GLP diagnosis was confirmed in 10 patients, with a median of three needle passes. The remaining four patients were laparoscopically diagnosed with GLP. The diagnostic ability of EUS-FNA biopsy for GLP was 76.9%, and EUS-FNA biopsy contributed to GLP diagnosis in 18.5% (10/54) of all cases. None of the 13 patients exhibited EUS-FNA biopsy-related adverse events. Univariable and multivariable analyses revealed an absence of superficial ulcerations as a predictor of false-negative endoscopic biopsy findings in patients with GLP. These results suggest EUS-FNA biopsy as a minimally invasive and safe alternative diagnostic modality for GLP in cases where conventional endoscopic biopsy fails to verify malignancy, although prospective studies with larger cohorts are warranted to confirm these findings.

## 1. Introduction

Gastric cancer (GC), the fifth most common cancer and the fourth leading cause of cancer-related deaths worldwide [1], exhibits morphological, biological, and clinical heterogeneity and is classified into several subtypes [2,3,4,5]. Gastric linitis plastica (GLP) is a distinct subtype of GC characterized by segmental or diffuse thickening and rigidity of the gastric wall that results in a leather bottle-like appearance, as demonstrated by imaging studies [3,6,7]. Histologically, poorly cohesive cells and/or signet-ring cells that can diffusely infiltrate the gastric wall characterize GLP [6,7,8]. Epidemiologically, GLP accounts for 7–10% of all primary GCs, with its frequency increasing in recent years [9]. Importantly, GLP is associated with poor prognosis because of both advanced stage diagnosis and a low curative resection rate [10,11,12]. The diagnostic criteria for GLP have been recently established amid the long-standing lack of consensus on GLP definition. Prior to this, terms such as diffuse infiltrating gastric cancer, scirrhous gastric cancer, and Borrmann type 4 gastric cancer have been indiscriminately used to define GLP. The recently developed pretherapeutic diagnostic scoring criteria for GLP—the Saint Louis linitis score, which exhibits high sensitivity and specificity [13]—is expected to aid in discriminating GLP from other GC subtypes. 

Histopathological examination reveals that the gastric mucosa is often spared from tumor cell invasion given that GLP arises from the submucosa. Therefore, conventional superficial endoscopic biopsy sampling is negative for tumor cells in a significant fraction of GLP cases [14,15], hindering its diagnosis. Since missed diagnosis adversely affects the treatment and prognosis of patients, the development of diagnostic modalities other than the conventional endoscopic biopsy is warranted. To date, several alternative diagnostic techniques, such as percutaneous fine-needle aspiration (FNA) biopsy [16,17], endoscopic mucosal resection (EMR) [18,19], endoscopic submucosal dissection (ESD) [20], and endoscopic ultrasound (EUS)-guided FNA biopsy [21,22,23] have been described in case reports and a small number of case series. However, there is no universal consensus on alternative approaches for GLP diagnosis in patients with negative standard endoscopic biopsy results. Among the available alternative diagnostic approaches for GLP, EUS-FNA biopsy has been increasingly adopted as a minimally invasive diagnostic modality for lesions in and adjacent to the gastrointestinal tract [24]. Theoretically, EUS-FNA may be advantageous for the diagnosis of GLP infiltrating the submucosa, since tissue samples from deep lesions including those in the submucosa can be acquired with this approach. However, limited data are available regarding the diagnosis of GLP by EUS-FNA biopsy [21,22,23]. The present study aimed to investigate the value of EUS for GLP diagnosis by evaluating outcomes of EUS-FNA biopsy in patients with GLP and negative endoscopic biopsy findings.

## 2. Materials and Methods

### 2.1. Patient Eligibility and Study Design

This was a single-center retrospective study conducted at Kindai University Hospital between June 1, 2016 and December 31, 2020. Patients diagnosed with GLP were identified using electronic medical records and a prospectively accumulated endoscopic database. Specifically, cases were searched using the terms “gastric linitis plastica”, “diffuse infiltrating gastric cancer”, “scirrhous gastric cancer”, and “Borrmann type 4 gastric cancer”. Candidate cases were reclassified using the recently published Saint Louis linitis score [13]. The scoring system consists of six variables including endoscopic, EUS, and biopsy findings. There were three variables corresponding to endoscopic findings, including the presence of large folds and/or gastric thickening on at least one segment (1.5 points), pangastric infiltration (2 points), and the presence of gastric stenosis (1 point). Next, there were two variables corresponding to EUS findings, including a circumferential thickening on at least one segment (0.5 points) and a predominance of the lesion on the third hyperechoic layer (1 point). Finally, there was one variable on the histological report on endoscopic biopsies, which was the presence of poorly cohesive cells and/or signet ring cells (1.5 points). The diagnostic performance of this scoring system was assessed by a receiver operating characteristic (ROC) curve analysis. This demonstrated that, when the area under the ROC curve value is 0.967 with a sensitivity of 94% and a specificity of 89%, the total score of all six variables was 2.75 as a threshold [13]. Based on this finding, cases with a Saint Louis linitis score of ≥ 3 were considered to have GLP, and were included in the present study. The exclusion criteria included patients with suspicious secondary linitis plastica due to the metastasis of a cancer other than GC. The study protocol was approved by the Institutional Review Board of Kindai University and conducted in accordance with the Helsinki Declaration.

### 2.2. Endoscopic Procedures

Ordinary superficial endoscopic biopsy procedures were performed in all cases with suspicious GLP. In the present study, Radial Jaw™ 4 standard-capacity biopsy forceps with a cup outer diameter of 2.2 mm (Boston Scientific, Natick, MA, USA) were used. Endoscopic biopsies were performed at multiple locations, and included the margin of ulcerations if present. EUS was performed using a linear GF-UCT260 echoendoscope (Olympus Medical Systems, Tokyo, Japan) in patients with negative endoscopic biopsy findings. Endosonographic images were observed using an ALOKA ProSound F75 or an ALOKA ARIETTA 850 processor (Hitachi Aloka Medical, Tokyo, Japan). The gastric wall was observed under EUS guidance with reference to computed tomography and endoscopic images, and the status of the thickened gastric wall was evaluated based on size, echo intensity, and layer structure. Subsequently, EUS-FNA biopsy samples were collected from the thickened gastric wall for histological evaluation. All EUS-FNA biopsy procedures were performed by experienced endoscopists (K.M., K.K., and K.Y.). A 22-gauge FNA needle (EZ Shot 3 Plus, Olympus Medical Systems) was used until July 2018, whereas a 22-gauge Franseen-tip core biopsy needle (Acquire, Boston Scientific)—also known as a fine-needle biopsy (FNB) needle—was used starting in July 2018. Briefly, the thickened gastric wall was punctured, the inner stylet was removed, and a 20-mL syringe was attached to the needle hub. Next, suction was applied while moving the needle within the lesion for 10–15 to-and-fro strokes. The needle was withdrawn after each pass, and the obtained sample was expressed onto a slide using a syringe filled with saline. Samples were processed for histological analysis. Rapid onsite evaluation (ROSE) was not performed. Instead, obtained samples were immediately and carefully checked by the endoscopist for the presence of a visible yellowish-white tissue core. At least three passes were performed per patient. If the presence of the tissue core could be confirmed after the third needle pass, the puncture was terminated, whereas if it could not be confirmed, another puncture was added. A maximum of five needle passes were allowed per patient. 

### 2.3. Outcome Measures

The primary outcome was the proportion of patients who received a histological diagnosis of poorly cohesive tumor cells based on EUS-FNA biopsy among those in whom endoscopic biopsy failed to verify malignancy. The secondary outcomes included the modalities used to reach the final pathological diagnosis, the diagnostic ability of EUS-FNA biopsy, the procedural details, and any adverse events (AEs) related to EUS-FNA biopsy. The factors associated with negative endoscopic biopsy findings in patients with GLP were evaluated using a variety of parameters in univariable and multivariable analyses. The medical records of patients who were followed until the end of the study (30 June 2021) or death were reviewed. 

### 2.4. Statistical Analysis

Data were presented as numbers with percentages for categorical variables and medians with interquartile ranges (IQRs) for continuous variables. Statistical comparisons were performed using Fisher’s exact test for binomial data. Univariable and multivariable logistic regression analyses were performed to explore factors associated with false-negative endoscopic biopsy findings in patients with GLP, and to determine odds ratios (ORs) with 95% confidence intervals (CIs). To assess the diagnostic ability of specific continuous variables, cutoffs were set to median values for all patients. The cutoff serum carcinoembryonic antigen (CEA) concentration was set to the upper limit of the normal value of 5.0 ng/mL. Factors with a *p* value of < 0.1 in the univariable analysis were included in the subsequent multivariable analysis. All statistical analyses were performed using GraphPad Prism 9.0 (GraphPad Software, La Jolla, CA, USA). *p* values < 0.05 were considered to indicate statistical significance.

## 3. Results

### 3.1. Selection of Patients with GLP

During the study period, the search of the electronic medical records and endoscopic databases using the study terms identified a total of 68 patients with suspicious GLP. Twelve patients with a Saint Louis linitis score of <3 were excluded. In addition, two patients who were considered to harbor secondary linitis plastica due to breast and colorectal advanced cancer, respectively, were excluded. Therefore, the final study cohort included 54 patients diagnosed with primary GLP. 

### 3.2. Characteristics of Patients with GLP

The baseline characteristics of 54 patients with GLP are presented in Table 1. The median patient age was 66 (IQR, 50–72) years, and there were 25 (46.3%) female patients. At diagnosis, 94.4% (51/54) of the patients were classified as clinical stage IV according to the 8^th^ UICC classification of gastric cancer [25]. Additionally, diffuse and segmental gastric wall thickening was observed in 57.4% (31/54) and 42.6% (23/54) of the patients, respectively. Serum CEA concentration was elevated in 27.8% (15/54) of the patients (normal range, ≤5.0 ng/mL). The median Saint Louis linitis score was 5 (IQR, 4–5). Superficial ulcerations were present during endoscopic examination in 44.4% (24/54) of the patients. 

### 3.3. Utility of EUS-FNA Biopsy for Patients with GLP

A flow chart of modalities used for the histological diagnosis of GLP in the present study is depicted in Figure 1. Poorly cohesive cells and/or signet-ring cells were confirmed by endoscopic biopsy in 74.1% (40/54) of the patients, with a median of six biopsy samples collected per patient. EUS was performed in 13 of the remaining 14 patients with negative endoscopic biopsy findings, after the exclusion of one patient who immediately underwent laparoscopic examination (Figure 1). On EUS, the thickened gastric wall was visualized as a hypoechoic region with a loss of the normal five-layer wall structure (Figure 2). The median thickness of the gastric wall measured by EUS was 20 mm (Table 2). EUS-FNA biopsy was performed in all 13 patients, with a median of three needle passes per patient. The 22-gauge FNA and FNB needles were used in 3 and 10 patients, respectively (Table 2). The histological evaluation confirmed the diagnosis of poorly cohesive malignant tumor cells by EUS-FNA biopsy in 76.9% (10/13) of the patients (Table 2). Thus, EUS-FNA biopsy contributed to the final histological GLP diagnosis in 18.5% (10/54) of the patients in this study. Compared with a puncture needle, the diagnostic ability for malignancy was 33.3% with an EUS-FNA needle and 90% with an EUS-FNB needle (*p* = 0.108). None of the 13 patients exhibited any AEs related to EUS-FNA biopsy. 

### 3.4. Factors Associated with Negative Conventional Biopsy Findings in Patients with GLP

Finally, we elucidated factors predicting negative endoscopic biopsy findings in patients with GLP by comparing the 14 patients with negative endoscopic biopsy findings with the remaining 40 patients who received a GLP diagnosis based on the endoscopic biopsy findings in the present study (Table 3). A univariable analysis revealed that segmental GLP (OR 3.34, 95% CI 0.97–12.8, *p* = 0.063), absence of superficial ulcerations (OR 30.3, 95% CI 5.17–582.9, *p* = 0.002), and a Saint Louis linitis score <5 (OR 3.75, 95% CI 1.06–15.64, *p* = 0.049) were candidate factors associated with false-negative endoscopic biopsy findings in patients with GLP. Thus, these three variables were included in the multivariable logistic regression analysis, which revealed that the absence of ulcerations was a significant predictive factor of false-negative endoscopic biopsy findings in patients with GLP (OR 30.1, 95% CI 4.72–666.0, *p* = 0.003) (Table 3). Overall, these data suggested that EUS-FNA biopsy should be considered in patients with endoscopic findings highly suggestive of GLP in the absence of ulcerations. 

## 4. Discussion

In the present study, we evaluated the diagnostic value of EUS-FNA biopsy in GLP as an alternative modality in patients with inconclusive findings by standard endoscopic biopsy. We demonstrated the diagnostic ability of EUS-FNA biopsy in the absence of any AEs in more than three-quarters of the patients with GLP. In the overall cohort, EUS-FNA biopsy contributed to the histological diagnosis of GLP in approximately 20% of patients. To the best of our knowledge, this is the first cohort study to clarify the diagnostic value of EUS-FNA biopsy in GLP, and suggests that EUS-FNA biopsy might contribute to the accurate diagnosis of GLP in cases where standard endoscopic biopsy fails to confirm malignancy. Therefore, these results provide important clinical evidence for the diagnosis and management of patients with suspicious GLP.

Superficial endoscopic biopsy is the irrefutable standard diagnostic modality for GC; however, endoscopic biopsy has been reported to fail malignancy verification in up to one-third of GLP cases [14]. Rebiopsy with careful targeting is recommended in patients with false-negative endoscopic biopsy results among those with endoscopic findings highly suggestive of GLP. In the present study, endoscopic rebiopsy was performed in 71.4% (10/14) of inconclusive cases after the first endoscopic biopsy. However, the second endoscopic biopsy results did not provide additional evidence of malignancy in any of the cases. Since superficial ulcerations were noted in only one patient, the target biopsy site could not be identified by endoscopy. In addition, more than half of the cases (8/14) exhibited segmental gastric wall thickness accompanied by stenosis, hindering the ability to biopsy from appropriate sites that displayed mucosal tumor invasion. Therefore, reaching a malignancy diagnosis may be challenging despite repeated biopsies in some cases. One study recommended three to four biopsies to be collected from viable tissue for correct pathological diagnosis of GCs, and reported that further biopsies did not increase the rate of positive diagnosis [26]. A median of six biopsies were performed in the present study; therefore, utilizing other diagnostic techniques rather than increasing the number of biopsies might be prudent in patients with inconclusive diagnosis based on endoscopic biopsy.

To date, several tissue sampling methods have been reported as alternatives to conventional endoscopic biopsy yielding nondiagnostic material. Zhou et al. developed a deep and large biopsy technique for the diagnosis of gastric infiltrating tumors with negative malignant endoscopic biopsy findings [19]. The authors performed bite-on-bite biopsies in the areas targeted by EMR after identifying the thickest site of the gastric wall via EUS, and demonstrated that the positive diagnostic rate of the deep and large biopsy technique was high (80.6%). However, 19.4% of the patients experienced bleeding as a procedure-associated AE. Chiyo et al. reported a patient with GLP who was successfully diagnosed with submucosal endoscopic sampling using ESD [20]. The authors performed submucosal biopsies after creating a submucosal tunnel with ESD, and cauterized several vessels visualized in the submucosa using hemostatic forceps to prevent bleeding after the submucosal biopsies. These biopsy techniques, which may be useful for deep tissue sampling when utilized in combination with EMR or ESD, nonetheless harbor non-negligible risk for bleeding or perforation.

Since its first description three decades ago, EUS-FNA biopsy has been widely used for deep tissue sampling [24,27]. EUS-FNA biopsy is considered as a very safe diagnostic modality with a morbidity rate of <1% [28]. During EUS-FNA biopsy, intervening vessels in the puncture site can be clearly visualized using color Doppler, which markedly reduces the risk of bleeding. According to a recent systematic review including 10,941 patients, the overall bleeding and perforation rates were 0.13% and 0.02%, respectively [28], which were markedly lower than those for submucosal sampling techniques used in combination with EMR or ESD as described above. Among long-term AEs associated with percutaneous FNA and EUS-FNA biopsy, needle tract seeding is a serious AE that affects prognosis [29,30]. Theoretically, the risk of needle tract seeding is estimated to be lower in EUS-FNA biopsy than in percutaneous FNA because the puncture needle is moved within the gastric wall using to-and-fro strokes during EUS-FNA. However, since the thickness of the gastric wall in patients with GLP is usually less than 2 cm, we must be aware of the potential risk for penetrating the gastric wall during EUS-FNA, which could make tumor cell seeding into the peritoneum. EUS is also useful for detecting a small amount of ascites, which suggests the presence of peritoneal dissemination and can change the further treatment modality. A previous retrospective study evaluating 101 patients who underwent EUS-guided paracentesis has shown the sensitivity, specificity, and diagnostic accuracy of ascites sampling at 80%, 100%, and 96%, respectively [31]. However, in the present study, because none of the 13 patients who underwent EUS-FNA biopsy for the thickened gastric wall exhibited perigastric ascites during EUS, EUS-guided paracentesis was not performed. If a small amount of ascites can be visualized by EUS, the diagnosis of GLP using EUS can be improved by sample collecting not only from the gastric wall but also from the ascites.

One limitation of EUS-FNA biopsy is the small specimen size with scant cellularity and lack of histologic architecture, which restricts the complete analysis of tumor tissue for diagnosis, especially in patients with GLP. This is evidenced by the low diagnostic accuracy rate (54%, 13/24) of EUS-FNA using a standard FNA needle on the thickened gastric wall reported in a previous study [32]. Similarly, the diagnostic ability of EUS-FNA using a standard FNA needle was low (33%) in the present study. To overcome this limitation, we used a new FNB needle once it became available to obtain adequate core tissue specimens from the gastric wall, which allowed for the preservation of tissue architecture for histological analysis [33,34,35]. The diagnostic ability of EUS-FNA biopsy using a Franseen-tip FNB needle was 90% (9/10), and the histological analysis for grade differentiation could be achieved in all nine cases. Although there was no significant difference between the two puncture needle types because of the small number of cases included in the present study, EUS-FNA biopsy using the FNB needle rather than the standard FNA needle should be recommended for the histological diagnosis of GLP. 

Regarding the need for pretherapeutic histological evidence, Song et al. proposed that diagnostic laparotomy under laparoscopy and even radical gastrectomy might be considered in patients with highly suspicious GLP based on multiple imaging studies [15]. The authors suggested that such a decision could be made based on patient consent and discussion by a multidisciplinary team, and that the verification of malignancy using biopsy samples was not required [15]. However, as indicated in the present study, more than 90% of the patients were contraindicated for upfront surgical resection with curative intent. Therefore, in such cases, it is desirable to initiate chemotherapy immediately after the confirmation of histological diagnosis. In addition, diseases that present with a diffuse or segmental thickened gastric wall that require differentiation from GLP include malignant lymphoma and benign etiologies such as gastritis secondary to *Helicobacter pylori* infection, Ménétrier disease, gastric amyloidosis, gastric sarcoidosis, gastroduodenal Crohn disease, and gastric immunoglobulin G4-related disease [23,36,37], all of which do not require surgery. Contrary to that proposed by Song et al. [15], core tissue sampling has recently become feasible due to the development of FNB needles, and EUS-FNA biopsy is expected to be adopted as a minimally invasive diagnostic modality. EUS-FNA biopsy can prevent unnecessary surgery, thereby allowing prompt procession to appropriate treatment in patients with inconclusive results by conventional endoscopic biopsy.

This study has several limitations. Although a prospectively accumulated endoscopic database was used, this was a retrospective study conducted at a single center, and thus selection bias is inevitable. Regarding patient enrollment, due to the lack of a clear definition for GLP, we used the recently introduced pretherapeutic diagnostic score for GLP to identify a homogenous group of patients. However, the validity of this diagnostic scoring system for GLP has not been well established, and future studies are warranted to confirm its utility. In terms of sample management, ROSE was not performed in this study. According to a recent meta-analysis, ROSE was associated with up to 3.5% improvement in diagnostic adequacy rates for EUS-FNA biopsy [38]. Therefore, if ROSE could be available in this study, the diagnostic ability of EUS-FNA biopsy for GLP may have been further improved. In addition, the study sample size was relatively small. Thus, the superiority of an EUS-FNB needle over an EUS-FNA needle for the histological diagnosis of GLP could not be demonstrated. Therefore, a prospective, multicenter study with a larger cohort is necessary to confirm the superiority of the EUS-FNB needle.

## 5. Conclusions

The present study suggests that EUS-FNA biopsy should be considered as a minimally invasive and safe alternative diagnostic modality for GLP in cases where conventional endoscopic biopsy fails to verify malignancy, especially in those without superficial ulcerations. Further prospective studies with larger cohorts are warranted to confirm these findings.

## Figures and Tables

**Figure 1 jcm-10-03716-f001:**
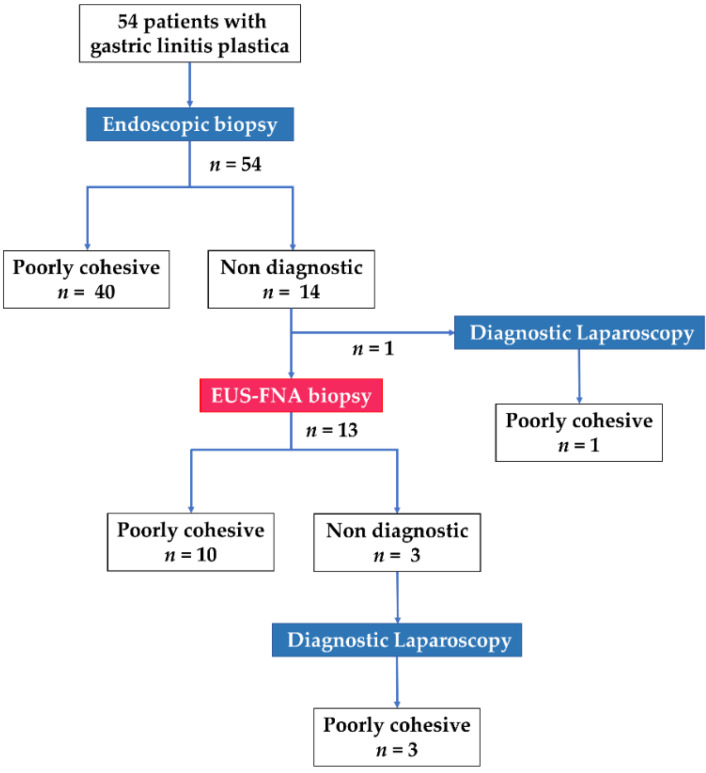
Flow chart of modalities used for histological diagnosis of gastric linitis plastica and specific diagnoses in 54 patients included in the study.

**Figure 2 jcm-10-03716-f002:**
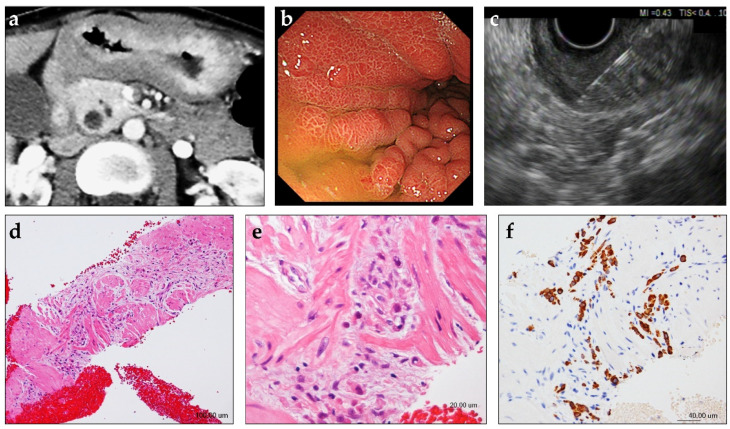
Case presentation of a patient with gastric linitis plastica in whom EUS-guided fine-needle biopsy (EUS-FNB) was useful for histological tissue sampling. A female patient in her 40s with abdominal pain and weight loss was referred to our hospital. (**a**) Contrast-enhanced computed tomography shows circumferential thickening of the gastric wall. (**b**) Upper endoscopy shows diffuse gastric wall thickening and large gastric folds. No ulceration is observed. Evaluation of multiple biopsies collected from the thickened gastric wall did not reveal evidence of malignancy. (**c**) On EUS, the thickened gastric wall is visualized as a 20-mm-thick, diffusely hypoechoic area with loss of normal layer definition. EUS-FNB of the thickened gastric wall was performed using a 22-gauge Franseen-tip core biopsy needle. (**d**,**e**) Histological diagnosis of the specimen collected by EUS-FNB showing a cluster of poorly differentiated adenocarcinoma within the muscularis propria of the gastric wall (hematoxylin/eosin staining). (**f**) Additional immunohistochemical analyses reveal that the tumor cells are positive for keratin AE1/AE3.

**Table 1 jcm-10-03716-t001:** Baseline characteristics of patients with gastric linitis plastica.

Patient Characteristics	*n* = 54
Median age, years (IQR)	66 (50–72)
Sex, male/female, *n* (%)	29 (53.7)/25 (46.3)
Cancer stage (UICC 8th classification)	
Ib/IIIb/IV, *n* (%)	1 (1.9)/2 (3.7)/51 (94.4)
Affected part of the stomach	
Diffuse/segmental, *n* (%)	31 (57.4)/23 (42.6)
Median serum CEA, ng/mL (IQR)	2.75 (1.7–5.55)
Ulcerations	
Yes/No, *n* (%)	24 (44.4)/30 (55.6)
Saint Louis linitis score	
3/3.5/4/5/6, *n* (%)	9 (16.7)/2 (3.7)/15 (27.8)/26 (48.1)/2 (3.7)
Median number of biopsies (IQR)	6 (5–9)
Endoscopic biopsy result	
Poorly cohesive/no malignancy, *n* (%)	40 (74.1)/14 (25.9)
Treatment, *n* (%)	
Chemotherapy	29 (53.7)
Surgical resection	1 (1.9)
Surgical resection after chemotherapy	16 (29.6)
Best supportive care	8 (14.8)

CEA, carcinoembryonic antigen; IQR, interquartile range; UICC, the International Union Against Cancer.

**Table 2 jcm-10-03716-t002:** Characteristics and outcomes of patients with GLP who underwent EUS-guided fine-needle aspiration biopsy following negative endoscopic biopsy (*n* = 13).

Age, Sex	CEA, ng/mL	Affected Part of the Stomach	Endoscopy		EUS			
			Ulceration	Number of Biopsies	Wall Thickness, mm	Type of Needle	Number of Needle Passes	EUS-FNA Biopsy Result
40, F	1.0	Diffuse	No	17	20	22-gauge, FNA	3	Atypical cells
81, M	5.4	Segmental	No	10	25	22-gauge, FNA	3	Poorly cohesive
83, M	2.4	Diffuse	No	8	20	22-gauge, FNA	3	No malignancy
53, M	1.4	Diffuse	No	13	15	22-gauge, FNB	5	Poorly cohesive
82, M	6.9	Segmental	Yes	6	25	22-gauge, FNB	3	Poorly cohesive
75, F	6.6	Segmental	No	7	15	22-gauge, FNB	3	Poorly cohesive
60, F	4.6	Segmental	No	5	15	22-gauge, FNB	3	Poorly cohesive
84, M	7.4	Segmental	No	16	20	22-gauge, FNB	3	Poorly cohesive
81, M	4.7	Segmental	No	6	15	22-gauge, FNB	3	Poorly cohesive
42, F	1.7	Diffuse	No	13	20	22-gauge, FNB	3	Poorly cohesive
68, M	2.3	Diffuse	No	22	20	22-gauge, FNB	3	Poorly cohesive
60, M	7.5	Segmental	No	9	18	22-gauge, FNB	5	Poorly cohesive
67, M	2.8	Segmental	No	9	20	22-gauge, FNB	3	No malignancy

CEA, carcinoembryonic antigen; F, female; FNA, fine-needle aspiration; FNB, fine-needle biopsy; GLP, gastric linitis plastica; M, male.

**Table 3 jcm-10-03716-t003:** Univariable and multivariable analyses of factors associated with negative endoscopic biopsy in patients with GLP.

	Univariable Analysis			Multivariable Analysis		
Variables	OR	95% CI	*p* Value	OR	95% CI	*p* Value
Age ≥ 67 years at diagnosis	1.33	0.39–4.72	0.646			
Sex						
Male	Reference					
Female	0.56	0.15–1.90	0.359			
Affected part of the stomach						
Diffuse	Reference			Reference		
Segmental	3.34	0.97–12.8	0.063	0.43	0.01–17.6	0.642
Ulcerations						
Yes	Reference			Reference		
No	30.33	5.17–582.9	**0.002**	30.13	4.72–666.0	**0.003**
Serum CEA						
≤5	Reference					
>5	1.67	0.43–6.11	0.443			
Number of biopsies						
<6	Reference					
≥6	3.14	0.83–15.5	0.115			
Saint Louis linitis score						
≥5	Reference			Reference		
<5	3.75	1.06–15.64	**0.049**	6.30	0.15–306.6	0.316

CEA, carcinoembryonic antigen; CI, confidence interval; GLP, gastric linitis plastica; OR, odds ratio. Bold indicates that the *p* values are statistically significant.

## Data Availability

The data presented in this study are available on request from the corresponding author.

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
