# Peer review of "Diagnostic Value of EUS-Guided Fine-Needle Aspiration Biopsy for Gastric Linitis Plastica with Negative Endoscopic Biopsy"

_jcm, 2021, doi:10.3390/jcm10163716_

Round 1

Reviewer 1 Report

Authors describe the efficacy of EUS-FNB for GLP with negative endoscopic biopsy. This article is well-analyzed and summarized. And also, this result provides us an important information in the management of GCP.

However, I would like to suggest some issues of this article with several comments and criticisms as following.

Major.

  1. What is the definition of sufficient material for completing the puncture?

The presence of whitish tissue?

The histological finding of GLP often exhibits massive submucosal fibrotic tissues with scattering cancer cells. Thus, small specimens obtained from FNB may lead to sampling errors. In this view, the judgement is important.

  1. Does rapid onsite evaluation (ROSE) work to diminish the sampling errors?

  1. What is the other diseases to differentiate GLP?

Authors need to discuss the differentiated diagnosis.

Author Response

Reviewer 1

Authors describe the efficacy of EUS-FNB for GLP with negative endoscopic biopsy. This article is well-analyzed and summarized. And also, this result provides us an important information in the management of GCP. However, I would like to suggest some issues of this article with several comments and criticisms as following.

A. We appreciate the general positive evaluation of our manuscript.

Major.

1. What is the definition of sufficient material for completing the puncture? The presence of whitish tissue? The histological finding of GLP often exhibits massive submucosal fibrotic tissues with scattering cancer cells. Thus, small specimens obtained from FNB may lead to sampling errors. In this view, the judgement is important.

A. Thank you for your very important and helpful comments. As described in the Materials and Methods section, ROSE was not available in our hospital. As this reviewer pointed out, small specimens obtained from FNB may lead to sampling errors. To reduce this risk, specimens obtained by EUS-FNA biopsy was macroscopically assessed for the presence of yellowish-white tissue core. In this study, at least three needle passes were performed per patient regardless of the presence of the tissue core. If the presence of the visible tissue core could be confirmed after the third needle pass, the puncture was terminated, whereas if it could not be confirmed, another puncture was added. A maximum of five needle passes were allowed per patient. We have added this information in the Materials and Methods section.

2. Does rapid onsite evaluation (ROSE) work to diminish the sampling errors?

A. Thank you for your comments. I totally agree with this reviewer that ROSE is useful in diminishing sample errors during EUS-FNA; however, as mentioned in the submitted manuscript, ROSE was not available at our hospital since ROSE requires a pathologist for cytological assessment immediately after needle puncture. If ROSE could be performed, the diagnostic ability of EUS-FNA biopsy for GLP may have been further improved. We have added this point to the limitation.

3. What is the other diseases to differentiate GLP? Authors need to discuss the differentiated diagnosis.

A. Thank you for your suggestion. As this reviewer pointed out, we have added discussions on the other diseases that require differentiation from GLP. We have added the following sentence in the Discussion section; In addition, diseases that present with diffuse or segmental thickened gastric wall that require differentiation from GLP include malignant lymphoma and benign etiologies such as gastritis secondary to Helicobacter pylori infection, Ménétrier disease, gastric amyloidosis, gastric sarcoidosis, gastroduodenal Crohn disease, and gastric immuno-globulin G4-related disease [23,35,36], all of which do not require surgery.

Reviewer 2 Report

This is a retrospective study about EUS-FNAB for linitis plastica. Considering difficulty in pathologic diagnosis, this study has a clinical meaning, However, there are several issues to be more clarified.

(1) The authors used the Saint Louis linitis score. This score system was recently published in Gastric Cancer. Therefore, the score system should be explained in the Method section.

(2) Although the gastric wall is thick in the linitis plastica, the thickness is usually less than 2 cm. Therefore, during EUS-FNAB procedure, there is a risk for penetrating the gastric wall, which could make the cancer cell seeding into the peritoneum. Of course, most patients with gastric wall thickening are inoperable, but this situation is important for patients who will undergo surgical resection. Therefore, the caution and risk of peritoneal seeing should be stated in the discussion section.

(3) Besides EUS-FNAB, EUS is very helpful for detecting a small amount of ascites, which suggests the presence of peritoenal seeding and can change the further treatment modality. If possible, the data about this issues is recommended to be included in the main text.

Author Response

Reviewer 2

This is a retrospective study about EUS-FNAB for linitis plastica. Considering difficulty in pathologic diagnosis, this study has a clinical meaning, However, there are several issues to be more clarified.

A. We appreciate careful reading and positive comments of this reviewer.

1. The authors used the Saint Louis linitis score. This score system was recently published in Gastric Cancer. Therefore, the score system should be explained in the Method section.

A. Thank you for your suggestion. As you advised, we have added the details of the recently published scoring system in the Materials and Methods section.

2. Although the gastric wall is thick in the linitis plastica, the thickness is usually less than 2 cm. Therefore, during EUS-FNAB procedure, there is a risk for penetrating the gastric wall, which could make the cancer cell seeding into the peritoneum. Of course, most patients with gastric wall thickening are inoperable, but this situation is important for patients who will undergo surgical resection. Therefore, the caution and risk of peritoneal seeing should be stated in the discussion section.

A. We appreciate your insightful comments. I totally agree with your opinion on the potential risk of needle tract seeding during EUS-FNAB procedure. Based on your suggestion, we have modified the text as follows; Theoretically, the risk of needle tract seeding is estimated to be lower in EUS-FNA biopsy than in percutaneous FNA because the puncture needle is moved within the gastric wall using to-and-fro strokes during EUS-FNA. However, since the thickness of the gastric wall in patients with GLP is usually less than 2 cm, we must be aware of the potential risk for penetrating the gastric wall during EUS-FNA, which could make tumor cell seeding into the peritoneum.

3. Besides EUS-FNAB, EUS is very helpful for detecting a small amount of ascites, which suggests the presence of peritoenal seeding and can change the further treatment modality. If possible, the data about this issue is recommended to be included in the main text.

A. Thank you for your suggestion. As the authors pointed out, EUS is very useful for detecting a small amount of ascites, which suggests the presence of peritoneal dissemination. A previous retrospective study evaluating 101 patients who underwent EUS-guided paracentesis has shown the sensitivity, specificity, and diagnostic accuracy of ascites sampling at 80%, 100%, and 96%, respectively (Wardeh R, et al. Cancer Cytopathol. 2011; 119(1): 27-36). We have also reported a case where ascites sampling in the pelvis by transrectal EUS-FNA was useful for diagnosing recurrence of gastric cancer (Minaga K, et al. Dig Liver Dis. 2018; 50(3): 311). However, in the present study, because none of the 13 patients who underwent EUS-FNA biopsy for the thickened gastric wall exhibited perigastric ascites during EUS, ascites sampling was not performed. We have added this information in the main text.